# A Novel Ferroptosis-Related Gene Signature Predicts Overall Survival of Breast Cancer Patients

**DOI:** 10.3390/biology10020151

**Published:** 2021-02-14

**Authors:** Haifeng Li, Lu Li, Cong Xue, Riqing Huang, Anqi Hu, Xin An, Yanxia Shi

**Affiliations:** 1Department of Medical Oncology, Sun Yat-sen University Cancer Center, Guangzhou 510060, Guangdong, China; lihf@sysucc.org.cn (H.L.); lilu1@sysucc.org.cn (L.L.); xuecong@sysucc.org.cn (C.X.); huangrq@sysucc.org.cn (R.H.); huaq@sysucc.org.cn (A.H.); 2State Key Laboratory of Oncology in South China, Guangzhou 510060 Guangdong, China; 3Collaborative Innovation Center for Cancer Medicine, Guangzhou 510060, Guangdong, China

**Keywords:** breast cancer, ferroptosis, gene signature, prognosis, immune status

## Abstract

**Simple Summary:**

Ferroptosis is an iron-dependent cell death which is distinctive from common forms of cell death. Accumulating evidence indicated the close relationship between ferroptosis and numerous human diseases. Regarding breast cancer, a related study indicated that some targeted medicines could induce ferroptosis, furthermore, some basic research found that ferroptosis-related genes were closely related to breast cancer. However, the correlation between ferroptosis-related genes and breast cancer patients’ prognosis remains unknown. We built an 8-ferroptosis-related-gene model to predict breast cancer patients’ prognosis. This model could stratify patients into high- or low-risk groups. Additionally, tumor microenvironment analyses displayed differently enriched immune cells and immune pathways between these two groups. This 8-gene model is believed to be of great value in predicting prognosis for breast cancer patients.

**Abstract:**

Breast cancer is the second leading cause of death in women, thus a reliable prognostic model for overall survival (OS) in breast cancer is needed to improve treatment and care. Ferroptosis is an iron-dependent cell death. It is already known that siramesine and lapatinib could induce ferroptosis in breast cancer cells, and some ferroptosis-related genes were closely related with the outcomes of treatments regarding breast cancer. The relationship between these genes and the prognosis of OS remains unclear. The data of gene expression and related clinical information was downloaded from public databases. Based on the TCGA-BRCA cohort, an 8-gene prediction model was established with the least absolute shrinkage and selection operator (LASSO) cox regression, and this model was validated in patients from the METABRIC cohort. Based on the median risk score obtained from the 8-gene model, patients were stratified into high- or low-risk groups. Cox regression analyses identified that the risk score was an independent predictor for OS. The findings from CIBERSORT and ssGSEA presented noticeable differences in enrichment scores for immune cells and pathways between the abovementioned two risk groups. To sum up, this prediction model has potential to be widely applied in future clinical settings.

## 1. Introduction

Breast cancer, as a global health concern, is the most common malignancy among women and ranks as the second leading cause for cancer-related death in women. Additionally, breast cancer has become the second most frequently diagnosed cancer worldwide, and will be diagnosed in one woman out of eight [1,2]. In Asia, one in three women faces the risk of breast cancer during their lifetime [3]. The five-year survival rate for breast cancer patients with stages III and IV were 57% and 23.4%, respectively [4]. It is commonly known that multigene signatures could provide risk stratification and prognostic prediction in breast cancer, such as PAM50 signature [5]. Also, this kind of multigene signature brings insight to molecular biologic characteristics of breast cancer, as transcriptome or related molecular biologic data was the original source of constructing such a prognostic prediction model. Therefore, this study aims to develop a ferroptosis-related gene signature to predict overall survival (OS) for breast cancer patients.

Ferroptosis is an iron-dependent cell death discovered by Dixon et al. Iron-dependent reactive oxygen species (ROS) increase, cell membrane thickening and mitochondrial volume reduction are biologic characteristics of ferroptosis [6]. The accurate mechanism underlying ferroptosis sensitivity in tumor cells is uncertain, but it is already known that different cancer types possess different levels of sensitivity to ferroptosis. Concerning breast cancer, based on current studies, iron, ACSL4 (Acyl-CoA synthetase long chain family member 4), PUFAs (polyunsaturated fatty acids), GPX4 (glutathione peroxidase-4) and p53 have been identified as vital regulators in ferroptosis pathway, also promising treatment targets for breast cancer. Related studies reported that growth of breast cancer cells was dependent on iron, and the iron chelator Dp44mT exhibited antitumor effects in breast cancer [7,8]. ACSL4 participates in the process of ferroptosis via enriching cellular membranes with long polyunsaturated n-6 fatty acids [9]. Antitumor activity of PUFAs is achieved by upregulating ROS in breast cancer cells. Some breast cancer cells acquire the ability of drug resistance through GPX4, namely those which are vulnerable to ferroptosis induced by GPX4 inhibition [10], suggesting that the GPX4 inhibitor might become a potential agent to overcome drug resistance in breast cancer. P53, a well-known tumor suppressor gene, has been reported to positively regulate the process of erastin-induced ferroptosis in breast cancer cells [11]. A recent study reported that ferroptosis was more prone to be induced by siramesine and lapatinib rather than four common ferroptotic reagents (erastin, RSL3, ML210 and ML162), and demonstrated that lapatinib-induced ferroptosis was not via targeting EGFR and HER2, which imply there were other targets in the induction process of ferroptosis by lapatinib [6]. Some researchers have found that triple-negative breast cancer is more sensitive to ferroptosis than ER-positive breast cancer [9].

A previous study showed that some ferroptosis-related genes had the potential to be promising treatment targets in breast cancer, such as iron, ACSL4, GPX4, SLC7A11 and SLC3A [9,12]. Therefore, it is important to identify the relationship between ferroptosis and prognosis in breast cancer patients. Studies with regard to ferroptosis-related gene signature as a prognostic marker are emerging. Liu et al. reported that the prognostic signature of 19 ferroptosis-related genes for glioma exhibited potential as a biomarker of OS in glioma patients [13]. Kwon et al. constructed a nuclear receptor meta-pathway (NRM) model identifying the patients for therapeutic intervention using erastin, which has been extensively accepted as a new ferroptosis-related antitumor therapy [14]. A 10-ferroptosis-related gene signature established by Liang et al was able to predict the prognosis for patients with hepatocellular carcinoma [15]. In this study, we developed a ferroptosis-related gene signature-based prognostic model for breast cancer patients, and investigated the difference of tumor microenvironment immunity between different risk groups classified by this model.

## 2. Materials and Methods

### 2.1. TCGA-BRCA Cohort and METABRIC Cohort

The transcriptome data and clinical characteristics of breast cancer patients in TCGA-BRCA and METABRIC were obtained from GDC (https://protal.gdc.cancer.gov/repository) and cBioProtal (www.cbioportal.org/), respectively. Additional normal breast tissue mRNA data were obtained from GTEx (https://gtexportal.org/home/datasets). This study included 60 genes related to ferroptosis, and these genes are detailed in Appendix A.

### 2.2. Construction and Validation of a Novel Prognostic Ferroptosis-Related Gene Signature 

Firstly, the differently expressed genes (DEGs) between tumor and normal tissues in the TCGA-BRCA cohort were screened with “DESeq2” R package with the threshold of p.adj < 0.05 and |logFC| > 1, then, the ferroptosis-related genes with prognostic value were identified with univariate cox regression of OS. Then, the least absolute shrinkage and selection operator (LASSO) Cox regression was carried to build a prognostic model with the “glmnet” R package. The variables that were used to build the model were the non-zero coefficients under the minimum lambda condition. Risk score calculation was based on the expression level of normalized gene and regression coefficient of corresponding gene, which was as follows: risk score = sum (expression level of each gene*corresponding coefficient). Then, patients were grouped into high- or low-risk group based on the median risk score. The concordance index (c-index) assessing the predictive accuracy of this 8-gene model was obtained by “risksetROC” R package.

### 2.3. The Tumor Microenvironment Analysis

Cell type identification by estimating relative subsets of known RNA transcripts (CIBERSORT) tool was used to identify the immune cell fractions of 22 distinct leukocyte subsets. The differences of immune-related pathways between groups were evaluated through the single-sample gene set enrichment analysis (ssGSEA) achieved by R package “gsva”. The immune-related gene sets used are provided in Appendix A.

### 2.4. Statistical Analysis

The chi-squared test of Fisher’s exact test was applied to compare the difference in proportions between groups. The Kaplan–Meier curve and log-rank test were used to compare OS in different risk groups. The independent predictors for OS were identified via univariate and multivariate cox regression. All reported p-values correspond to two-sided tests, and p-values < 0.05 were considered statistically significant. All statistical analyses were carried out with R, version 3.6.3.

## 3. Results

In this study, TCGA-BRCA and METABRIC cohorts included 1043 and 1904 breast cancer patients, respectively. Table 1 details the baseline information of all the patients.

### 3.1. Identification of Prognostic Ferroptosis-Related Genes in the TCGA-BRCA Cohort

A total of 60 ferroptosis-related genes are shown in Figure 1a, and only 18 of them were differently expressed between tumor and normal tissues (Figure 1b). Out of the 60 ferroptosis-related genes, 9 were of prognostic value in breast cancer, of note, only ACSF2 presented as a favorable prognostic factor (Figure 1c). Genes which were not only expressed differently between tumor and normal tissue but also of prognostic value were SQLE, SLC7A11 and CHAC. In tumor tissue, these 3 genes were all upregulated, and higher expression predicted worse overall survival (OS) (Figure 1b,c). 

### 3.2. A Prognostic Model Construction in the TCGA-BRCA Cohort 

Based on above-mentioned 9 candidate genes, we preformed LASSO Cox regression analysis to construct the ferroptosis-related genes signature-based risk model. When the model reached the minimum of lambda (λ), a prognostic model with 8 non-zero coefficient genes (ALOX15, CHAC1, CISD1, CS, SLC7A11, EMC2, G6PD, ACSF2) was built (Figure A1), and the SQLE was removed for its zero coefficient. Median risk score classified patients into high-risk (n = 521) or low-risk (n = 522) groups, which was calculated as follows: risk score = 0.222 × expression level (EL) of ALOX15 + 0.081 × EL of CHAC1 + 0.182 × EL of CISD1 + 0.304 × EL of CS + 0.070 × EL of SLC7A11 + 0.230 × EL of EMC2 + 0.139 × EL of G6PD - 0.107 × EL of ACSF. Table 2 shows that in the TCGA-BRCA cohort, the high-risk group was not associated with higher tumor stage and menopausal status, but more Asian patients were in the high-risk group and more Caucasian patients were in the low-risk group. The Kaplan–Meier curve revealed better OS for patients from the low-risk group compared to high-risk group (Figure 2a), and the maximum value of c-index was 0.715 (Figure 2b). 

### 3.3. 8-Gene Signature Validation in the METABRIC Cohort 

To confirm the reliability of this 8-gene model, patients in the METABRIC cohort were stratified into high- or low-risk group based on the median value of risk score calculated as described above. The high-risk group in the METABRIC was related with higher pathologic grade of lymph nodes (Table 2). Additionally, the Kaplan–Meier curve also indicated an inferior OS for patients in the high-risk group (Figure 2c), and the maximum value of c-index was 0.590 (Figure 2d). 

### 3.4. Independent Prognostic Value of the 8-Gene Signature 

In the univariate Cox regression analysis, the low-risk group had better OS in both the TCGA-BRCA cohort and the METABRIC cohort (HR = 0.515, 95% CI = 0.369–0.720, *p* < 0.001; HR = 0.780, 95% CI = 0.693–0.878, *p* < 0.001 respectively) (Table 3). In the multivariate Cox regression analysis, the low-risk group also showed better OS in these two cohorts (HR = 0.473, 95% CI = 0.332–0.673, *p* < 0.001; HR = 0.829, 95% CI = 0.735–0.935, *p* < 0.01, respectively) (Table 3). Therefore, risk score was an independent prognostic factor for OS. Moreover, older age was an unfavorable factor for OS in both cohorts (TCGA-BRCA: HR = 1.035, 95% CI = 1.022–1.048, *p* < 0.001; METABRIC: HR = 1.054, 95% CI = 1.046–1.062, *p* < 0.001, respectively) (Table 3). For the patients in the METABRIC cohort, history of chemotherapy treatment and higher pathologic grade of lymph nodes both independently predicted worse OS (HR = 1.741, 95% CI = 1.450–2.091, *p* < 0.001; HR = 1.705, 95% CI = 1.487–1.955, *p* < 0.001, respectively) (Table 3).

### 3.5. The Tumor Microenvironment Analysis in the TCGA-BRCA and METABRIC Cohort 

Considering the ssGSEA results of the TCGA-BRCA cohort, the enrichment scores of mast cells, neutrophils and type II IFN response in the low-risk group were higher compared to the high-risk group (*p* < 0.05, Figure 3a). Mast cells and type II IFN response in the low-risk group had higher enrichment scores in both the TCGA-BRCA and the METABRIC cohorts (Figure 3a,b). The CIBERSORT results of TCGA-BRCA indicated the difference of fraction of distinct leukocyte subsets between two risk groups—the proportions of memory B cells, naïve CD4+ T cells, eosinophils and neutrophils were higher in the low-risk group (*p* < 0.05, Figure 4a); the fractions of activated CD4+ memory T cells in the high-risk group were higher than that of the low-risk group in both cohorts, and activated NK cells and monocytes had higher proportions in the high-risk group of the METABRIC cohort (Figure 4a,b). 

## 4. Discussion

In this study, the 60 ferroptosis-related genes were obtained from review articles and published experiments [16,17,18]. All of them were analyzed to identify prognostic genes. Eventually an 8-gene prognostic model was constructed to predict the prognosis for breast cancer patients, which was validated in an external cohort from METABRIC. The fluctuation of c-index between 0.58–0.71 in Figure 2b and 0.57–0.59 in Figure 2d indicated a moderate model in terms of predictive accuracy. As the c-index reaches its highest at long-term follow-up in both cohorts, this model was believed to be suited for predicting long-term survival. Meanwhile, c-index maintains a stable level in the early stage, suggesting its value in short-term prognosis.

Ferroptosis is an iron-dependent cell death which is morphologically, biochemically and genetically distinct from apoptosis, necrosis and autophagy [19,20]. This process is symbolized by an iron-dependent increase of reactive oxygen species (ROS), thickening of the cell membrane and diminished volume of mitochondria [6]. Several inducing factors of ferroptosis have been reported, such as erastin, sulfasalazine, RSL3 and cysteine starvation [21]. Meanwhile, the regulators of ferroptosis were investigated as well, for instance, GPX4, P53, HSPB1, CISD1, CHAC1, CARS, SLC7A11, TFR1 [22]. CARS and TFR1 promote positive feedback for ferroptosis, however, SLC7A11, HSPB1, NRF2 and GPX4 are responsible for negative feedback for ferroptosis [23]. In terms of breast cancer, S Ma et al. found that the combination of siramesine and lapatinib induces ferroptosis via decreasing expression of ferroportin (FPN) and increasing transferrin expression. Their findings may provide a new treatment modality for apoptotic resistant breast cancer cells [6]. The correlation between ferroptosis-related genes and breast cancer patients’ prognosis remains unknown. 

Results in this study revealed 18 differently expressed ferroptosis-related genes between breast cancer tissue and normal tissue, and 3 out of them were of prognostic value. The univariate cox regression eventually found 9 ferroptosis-related genes with prognostic value and, eventually, an 8-gene prognostic model (ALOX15, CHAC1, CISD1, CS, SLC7A11, EMC2, G6PD, ACSF2) was constructed. A previous study revealed that 3 ferroptosis-related genes (SLC7A11, G6PD, CISD1) in hepatocellular carcinoma were upregulated in tumor tissue, and their high expression correlated with a poor prognosis [15]. These 3 genes were statistically significant in predicting overall survival in this study for breast cancer, moreover, the expression of SLC7A11 was also upregulated in breast cancer. CISD1 (CDGSH iron sulfur domain), an iron-containing outer mitochondrial membrane protein, negatively regulates ferroptotic cancer cell death. Inhibition of CISD1 increased iron-mediated intramitochondrial lipid peroxidation, leading to erastin-induced ferroptosis [24]. ALOX15 (arachidonate lipoxygenase 15) is related to producing lipid-ROS in gastric cancer. Zhang H et al. have reported that cisplatin and paclitaxel could promote secretion of miR-522 from CAFs (cancer-associated fibroblasts) via the USP7/hnRNPA1pathway, leading to ALOX15 suppression in gastric cancer cells, resulting in a poor therapeutic effect since the decreased chemosensitivity [25]. G6PD (glucose-6-phosphate dehydrogenase), a key enzyme that generates NADPH to maintain reduced glutathione (GSH), is capable of cleaning reactive oxygen species (ROS). A few studies have revealed that upregulation of G6PD promotes cancer progression in several types of carcinoma. Chen X et al. found that high G6PD expression predicts poor prognosis in bladder cancer, on top of that, the higher the levels of G6PD, the higher the tumor stage [26]. Not only in bladder cancer, G6PD was also overexpressed in colorectal cancer (CRC) cells, and high expression correlated with poor prognosis and poor outcome of oxaliplatin-based first-line chemotherapy in patients with CRC [27]. SQLE (squalene epoxidase), commonly known as a key enzyme of cholesterol synthesis, can also induce EMT (epithelial-to-mesenchymal transition) via regulating miR-133b in esophageal squamous cell carcinoma [28]. SLC7A11 is a unit of the glutamate-cystine antiporter system-xc (system-xc), which is mainly responsible for reduction of cystine to cysteine. Cysteine is one of units forming glutathione, which is an antioxidant protecting cells from lipid oxidative damage and ferroptosis. A recent study claimed that the combination of immunotherapy and radiotherapy suppresses SLC7A11, thus this combinatorial treatment modality promotes ferroptosis in cancer cells [29]. ChaC glutathione-specific gamma-glutamyl cyclotransferase 1 (CHAC1) degradation of glutathione enhances cystine starvation-induced ferroptosis in triple-negative breast cancer cells via the GCN2-eIF2α-ATF4 pathway [30]. Whether the abovementioned 8 genes play a huge part in OS for breast cancer patients through ferroptosis remains to be illustrated.

The interrelation of ferroptosis and tumor immune microenvironment remains elusive. To investigate the tumor immune microenvironment of breast cancer thoughtfully, we did not only focus on the abundance of immune cells, but also pay attention to the activity of immune response pathways. Therefore, we performed CIBERSORT to quantify the proportions of immune cells and ssGSEA to assess the activity of immune response pathways. The findings from CIBERSORT and ssGSEA displayed differently enriched immune cells and immune pathways between the high- and low-risk groups—ssGSEA revealed the different cell state in terms of activity levels of immune pathways, and CIBERSORT showed different immune cell infiltration fractions of the 22 distinct leukocyte subsets. Comparing the tumor immune microenvironment between the high- and low-risk groups, the high-risk group had more immunosuppressive cells and higher activity of immunosuppressive pathways, such as higher proportions of macrophage and Treg in the high-risk group. This finding proves the reliability of our model. Thus, it is worth assuming that there is a close connection between immunity and ferroptosis. Concerning the enrichment scores in ssGSEA, aDCs, APC co-stimulation, inflammation promoting, macrophage, T cell co-inhibition, Th1 cells, Treg and type I IFN response were remarkably higher in the high-risk group in both the TCGA-BRCA and the METABRIC cohorts. Increased macrophages or Treg cells have been reported to be related to poor prognosis in hepatocellular cancer patients [31,32,33], these two immune cells likewise increased in breast cancer patients in the high-risk group according to our study. It has been reported that M1 macrophage was related to tumor regression and inhibition of tumor growth. M1/Th1 responses are correlated with the release of proinflammatory cytokines such as TNF and IFN-γ (i.e., type II IFN) [34]. In the present study, the M1 macrophage, Th1 cells and inflammation promoting indeed had statistically higher scores in the high-risk group from the TCGA-BRCA cohort. Interestingly, the high-risk group had more tumor-inhibiting related cells (M1 and Th1), so it is reasonable to presume high expression of ferroptosis-related genes may elicit antitumor innate immune response, including the M1 polarization. One possible reason for poor prognosis in the high-risk group with high scores of M1 may be the high M2/M1 ratio as M2 macrophages have been shown to exhibit tumor-promoting ability via inducing angiogenesis factors [35]. Antitumor activity of type II IFN response decreased in the high-risk group in both cohorts, which could be the reason for poor OS in the high-risk group.

Limitations of our study are as follows: first, only public databases were used in this study, therefore, the use of the prognostic model in a real clinical setting remains controversial. Second, this model only includes long-term survival as an indicator of prognosis, short-term treatment response validation should be included as well. Third, duo to the intrinsic nature of the public databases, meaningful clinical information like chemotherapy history, tumor stage and metastatic lymph nodes were unavailable from public resource, which may make TCGA-BRCA and METABRIC cohorts incomparable, though we have tried to minimize the risk by multivariate cox regression analyses. In addition, we eventually established an 8 ferroptosis-genes prognostic model rather than those 3 prognostic differently expressed genes, which may have compromised the synergic effect of other genes.

## 5. Conclusions

To sum up, a novel prognostic signature consisting of 8 ferroptosis-related genes was established in the present study. In both derivation and validation cohorts, this signature showed reliable ability in predicting overall survival and correlated with intratumor microenvironment for breast cancer. Further investigation is needed in future research. 

## Figures and Tables

**Figure 1 biology-10-00151-f001:**
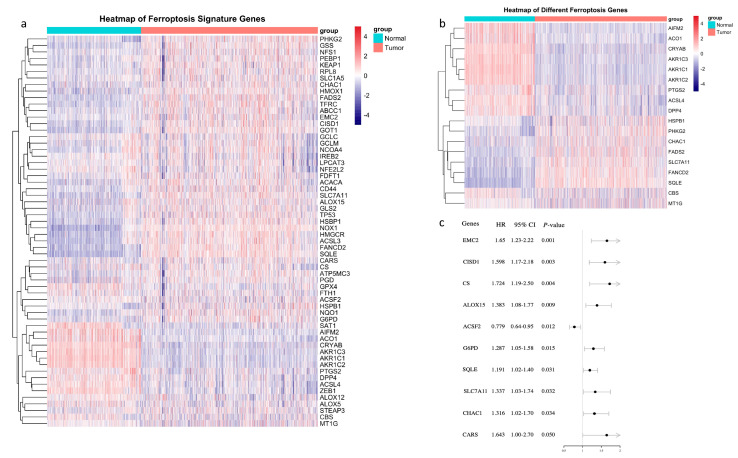
Identification of prognostic ferroptosis-related genes in the TCGA-BRCA cohort. (**a**) Heatmap of ferroptosis signature genes; (**b**) Eighteen genes were differently expressed between tumor and normal tissue; (**c**) Forest plots showed that nine ferroptosis-related genes were of prognostic value in breast cancer. HR, hazard ratio; 95% CI, 95% confidence interval.

**Figure 2 biology-10-00151-f002:**
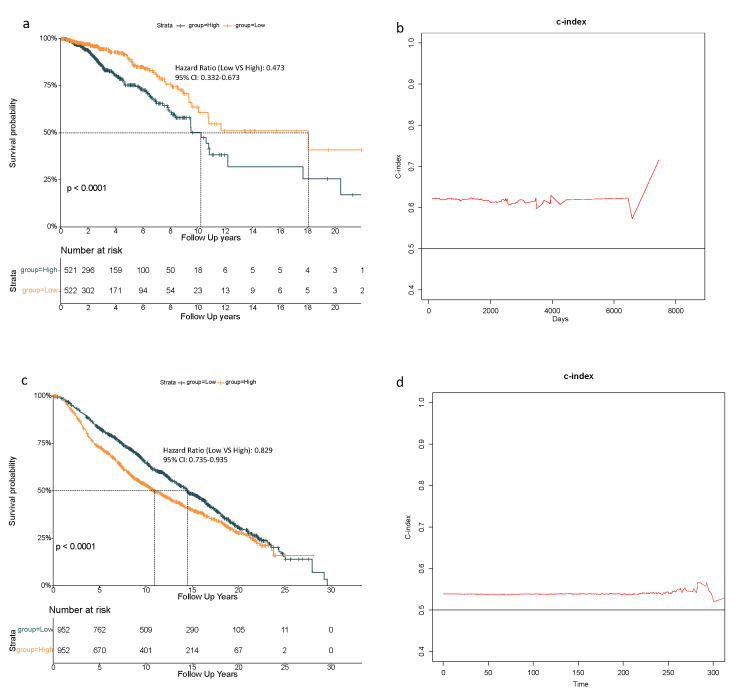
Overall survival analysis based on the 8-gene model in the TCGA-BRCA cohort. (**a**) Kaplan–Meier curves showed overall survival (OS) for high- and low-risk groups from the TCGA-BRCA cohort; (**b**) concordance index (c-index) assessed the predictive accuracy of this 8-gene model and the maximum value of c-index was 0. Validation in the METABRIC cohort. (**c**) Kaplan–Meier curves showed the overall survival (OS) for high- and low-risk groups from the METABRIC cohort; (**d**) concordance index (c-index) assessed the predictive accuracy of this 8-gene model and the maximum value of c-index was 0.590.

**Figure 3 biology-10-00151-f003:**
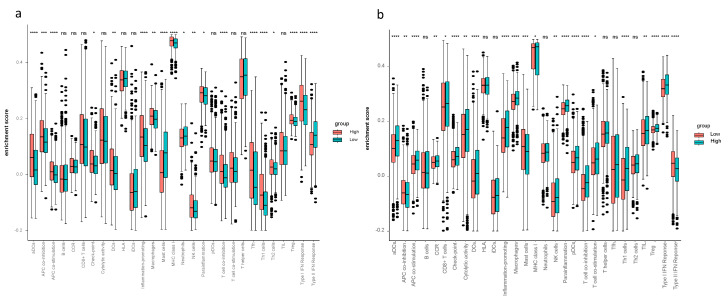
The single-sample geneset enrichment analysis (ssGSEA) in the TCGA-BRCA and METABRIC cohorts. The relative numerical values corresponding to the height of the histogram indicate different enrichment scores. (**a**) ssGSEA for the TCGA-BRCA cohort characterized different cell state in terms of activity levels of immune pathways between high- and low-risk groups. (**b**) ssGSEA for the METABRIC cohort characterized different cell state in terms of activity levels of immune pathways between high- and low-risk groups.

**Figure 4 biology-10-00151-f004:**
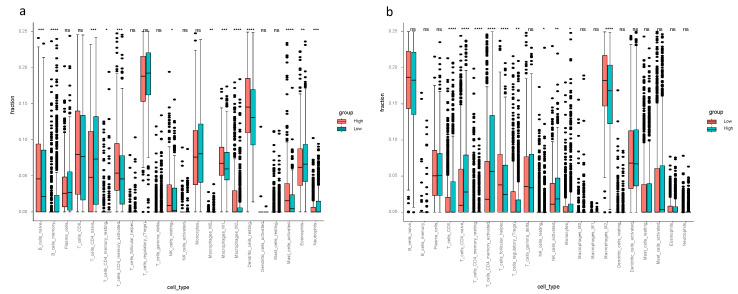
Cell type identification by estimating relative subsets of known RNA transcripts (CIBERSORT) in the TCGA-BRCA and METABRIC cohorts. The relative numerical values corresponding to the height of the histogram indicate different proportions. (**a**) CIBERSORT for TCGA-BRCA cohort showed the immune cell infiltration fraction of the 22 distinct leukocyte subsets. (**b**) CIBERSORT of METABRIC cohort showed the immune cell infiltration fraction of the 22 distinct leukocyte subsets.

**Table 1 biology-10-00151-t001:** Baseline characteristics of breast cancer patients included in this study.

Characteristics	TCGA-BRCA	METABRIC
N	1043	1904
**Age (mean (SD))**	58.33 (13.20)	61.09 (12.98)
**Gender (%)**		
Male	12 (1.2)	-
Female	1031 (98.8)	-
**Race (%)**		
Asian	57 (5.5)	-
Black	178 (17.1)	-
White	727 (69.7)	-
NA	81 (7.8)	-
**Stage (%)**		
I	178 (17.1)	-
II	589 (56.5)	-
III	234 (22.4)	-
IV	19 (1.8)	-
**Menopause status (%)**		
Pre	219 (21.0)	411 (21.6)
peri	72 (6.9)	0 (0)
Post	664 (63.7)	1493 (78.4)
NA	88 (8.4)	0 (0)
**Chemotherapy (%)**		
Yes	-	396 (20.8)
No	-	1508 (79.2)
**Pathologic N (%)**		
< = N1	-	1434 (75.3)
> = N2	-	470 (24.7)

**Table 2 biology-10-00151-t002:** Clinical characteristics in different risk groups.

Characteristics	TCGA-BRCA			METABRIC		
	High-Risk (n = 521)	Low-Risk (n = 522)	*p* Value	High-Risk (n = 952)	Low-Risk (n = 952)	*p* Value
**Age (mean (SD))**	58.34 (13.56)	58.32 (12.84)	0.983	60.66 (13.29)	61.51 (12.65)	0.151
**Gender (%)**			0.146			-
Male	9 (1.7)	3 (0.6)		-	-	
Female	512(98.3)	519(99.4)		-	-	
**Race (%)**			0.001			-
Black	94 (18.0)	84 (16.1)		-	-	
White	338 (64.9)	389 (74.5)		-	-	
Asian	38 (7.3)	19 (3.6)		-	-	
NA	51 (9.8)	30 (5.7)		-	-	
**Stage (%)**			0.401			-
I+II	376(72.2)	391(74.9)		-	-	
III+IV	135 (25.9)	118 (22.6)		-	-	
NA	10(1.9)	13(2.5)		-	-	
**Menopause status (%)**			0.201			0.373
Pre	117 (22.5)	102 (19.5)		214 (22.5)	197 (20.7)	
post	331 (63.5)	333 (63.8)		738(77.5)	755(79.3)	
peri	28 (5.4)	44 (8.4)		-	-	
NA	45 (8.6)	43 (8.2)		-	-	
**Chemotherapy (%)**			-			<0.001
Yes	-	-		265 (27.8)	131 (13.8)	
No	-	-		687 (72.2)	821 (86.2)	
**Pathologic N (%)**						0.017
< =N1				694 (72.9)	740 (77.7)	
> =N2				258 (27.1)	212 (22.3)	

**Table 3 biology-10-00151-t003:** Univariate and multivariate Cox regression analyses regarding OS in the TCGA-BRCA cohort and the METABRIC cohort.

Predictors	TCGA-BRCA METABRIC											
	Univariate Analysis			Multivariate Analysis			Univariate Analysis			Multivariate Analysis		
	HR	95% CI	*p*	HR	95% CI	*p*	HR	95% CI	*p*	HR	95% CI	*p*
**Age**	1.032	1.02-1.045	<0.001	1.035	1.022–1.048	<0.001	1.036	1.03–1.041	<0.001	1.054	1.046–1.062	<0.001
**Gender** **(Male vs. Female)**	0.84	0.1168–5.993	0.859	-	-	-	-	-	-	-	-	-
**Race**												
Black (Ref)												
White	0.826	0.552–1.236	0.353	-	-	-	-	-	-	-	-	-
Asian	0.617	0.188–2.029	0.427	-	-	-	-	-	-	-	-	-
**Stage** **(III/IV vs. I/II)**	2.665	1.905–3.730	<0.001	2.84	2.024–3.982	<0.001	-	-	-	-	-	-
**Menopause status**												
Pre (Ref)												
Post	1.279	0.842–1.943	0.248	-	-	-	1.685	1.431–1.983	<0.001	0.633	0.501–0.799	<0.001
Peri	0.906	0.457–1.794	0.776	-	-	-	-	-	-	-	-	-
**Group** **(Low vs. High)**	0.515	0.369–0.720	<0.001	0.473	0.332–0.673	<0.001	0.780	0.693–0.878	<0.001	0.829	0.735–0.935	<0.01
**Chemotherapy** **(Yes vs. No)**	-	-	-	-	-	-	1.228	1.057–1.427	0.007	1.741	1.450–2.091	<0.001
**Pathologic N** **(> = N2 vs. < = N1)**	-	-	-	-	-	-	1.952	1.714–2.222	<0.001	1.705	1.487–1.955	<0.001

## Data Availability

The data sets analyzed in this study are available on the public databases.

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
