# Peer review of "A Novel Ferroptosis-Related Gene Signature Predicts Overall Survival of Breast Cancer Patients"

_biology, 2021, doi:10.3390/biology10020151_

Round 1

Reviewer 1 Report

The authors created a survival-analysis (Cox regression) prognostic model of breast cancer overall survival using transcriptional (expression) levels of eight genes involved on ferroptosis, a type of iron-dependent programmed cell death. The rationale of the manuscript is sound, but the presentation of the Results and the Discussion can be improved.

Major suggestions

Discuss if missing data in the cohorts could have biased the model. For example, TCGA-BRCA had race information whereas METABRIC had not; METABRIC had information on chemotherapy, whereas the other not. Are the two cohorts really comparable?

Given C-index varied with follow-up time, discuss if the model would be best suited for short-term or long-term prognosis.

The authors selected 60 genes candidates due to their role in ferroptosis. They found that 18 were differentially expressed in tumour cells when compared to normal tissue. Among those, 9 had significant association with overall survival during univariate Cox regression and 3 had both differential expression and statistically significant Cox regression coefficients. However, the authors removed SQLE gene from the 9, ending with the 8 genes included in the model. It is not clear to me why SQLE gene was removed from the model, especially since the authors even mention this gene in the Discussion. Please clarify.

The authors performed tumor immunity microenvironment via CIBERSORT tool and ssGSEA. Please add a description how to interpret the results of Figure 3, highlighting how the two differ or complement one other. Please clarify what means the "score" (the Y-axis of Figure 3) given by both tools.

In the Conclusion, the authors wrote: "In addition, we eventually chose 9 prognostic ferroptosis-genes to establish prognostic model...". Should it be 8 genes?

Again in the Conclusion, the authors continue: "...rather than those 3 prognostics differently expressed genes, which was mainly because it is hard and worthless to build a prognostic model with 3 genes". What the authors mean by that? Worthless how? The cost-benefit ratio is worse? The model loses predictive power with less genes? Please clarify.

Minor suggestions

Improve quality of figures. Fonts are too small and they lose resolution when zooming in.

Again in Figure 3, it is best if panels a and c (and b and d as well) were grouped in a single panel, faceting by the cohorts. Since authors R, this can be easily achieved using ggplot2 faceting, for example. It will make much easier to compare the ssGSEA and CIBERSORT results side-by-side and not top-bottom.

Improve table formatting. The text in the tables were crammed (especially Table 3), so it was difficult to read.

Improve paper formatting. Several spaces in excess in the text and table cells.

Simple Summary: "Stacks of studies revealed...". Weird phrasing, I suggest a rewrite.

Reviewer 2 Report

Title: A Novel Ferroptosis-related Gene signature predicts overall survival of breast cancer patients

Summary: The authors identified ferroptosis-related genes, which could predict prognostic significance of breast cancer patients. As the authors described in simple summary, significance of ferroptosis in diverse disease has been widely studied recently. In this regard, defined ferrotpsis gene signature would be important. However, several unsolved questions still remain.

  1. To develop the concepts for the identification of ferroptosis gene signature, the ferroptosis pathway must play an important role or be augmented in tumors bu not in normal tissue. According to introduction, several reports demonstrated that ferroptsis is induced by siramesine and lapatinib. However, that doesn’t indicate enrichment of ferroptsis pathway in breast cancer patient. Furtheremore, among 60 genes, related to ferroptosis, only 18 genes are differentially expressed in tumor compared to normal tissue in figure 1. Therefore, I think the author need to first demonstrate the significance of ferrotosis pathway in breast cancer.
  2. More reference and description about ferroptosis-related gene signature as a prognostic marker (such as PMID:32733879, 32979793, 32760210) need to be included.
  3. How did authors define the 60 ferroptosis-related gene?
  4. What did numbers indicate for? More detail description should added in figure legend (figure 1C)
  5. Hazard ration should be included in figure 2A and 2C.
  6. What does the concordance index (c-index) mean? More detail explanation of Figure 2B and 2D should be included for better understanding.
  7. What does figure 3 imply for?? It should include a more detailed explanation of the notion and the concept of analysis, implication, etc.

Reviewer 3 Report

The authors describe in the manuscript a possible correlation among some gene upregulated in breast cancer patients and ferroptosis to predict cancer patients’ prognosis. 

Ferroptosis is a new form of cell death with a promising application in tumor therapy. To identify the type or the stage of the tumor more sensitive to ferroptois is actually a very interesting field of reseach.

The correlation of ferroptosis and tumor microenvironment immunity is another important point in this field and the authors try to shed light on some aspect of it.

Certainly, the work has some limitations, as highlighted by the authors ,and in the future the results will require experimental confirmation in cells (with overexpression and downregulation of the identified genes) and in mice; but the work was well done and the results discussed taking in consideration the data already published on other cancers.

Apart from minor text editing, the reviewer thinks that the manuscript could be published in the present form.

Author Response

Dear reviewer,

Really thank you  for your appreciation and positive encouragement. Your appreciation give us confidence to study and thinking.

Best wishes,

Yanxia Shi

Round 2

Reviewer 2 Report

The authors have tried to resolve all comment raised through the revision process. In this regard, this version of manuscript is suitable for publication in Biology.